

# A metatranscriptomic exploration of fungal and bacterial contributions to allochthonous leaf litter decomposition in the streambed

Aman Deep[1,2], Guido Sieber[2,3], Lisa Boden[2], Gwendoline M. David[4], Daria Baikova[5], Dominik Buchner[6], Jörn Starke[7], Tom L. Stach[3,7], Torben Reinders[2], Una Hadžiomerović[3,5], Sára Beszteri[2], Alexander J. Probst[3,7,8], Jens Boenigk[2,3] and Daniela Beisser[1,3]

[1] Department of Engineering and Natural Sciences, Westphalian University of Applied Sciences, Recklinghausen, North Rhine-Westphalia, Germany
[2] Biodiversity, Faculty of Biology, University of Duisburg-Essen, Essen, North Rhine-Westphalia, Germany
[3] Centre of Water and Environmental Research (ZWU), University of Duisburg-Essen, Essen, North Rhine-Westphalia, Germany
[4] Department of Plankton and Microbial Ecology, Leibniz Institute of Freshwater Ecology and Inland Fisheries, Stechlin, Brandenburg, Germany
[5] Aquatic Microbiology, Environmental Microbiology and Biotechnology, Faculty of Chemistry, University of Duisburg-Essen, Essen, North Rhine-Westphalia, Germany
[6] Aquatic Ecosystem Research, Faculty of Biology, University of Duisburg-Essen, Essen, North Rhine-Westphalia, Germany
[7] Environmental Metagenomics, Research Center One Health Ruhr of the University Alliance Ruhr, Faculty of Chemistry, University of Duisburg-Essen, Essen, North Rhine-Westphalia, Germany
[8] Centre of Medical Biotechnology (ZMB), University of Duisburg-Essen, Essen, North Rhine-Westphalia, Germany

Corresponding author
Aman Deep, aman.deep@uni-due.de

## ABSTRACT

The decomposition of organic matter is essential for sustaining the health of freshwater ecosystems by enabling nutrient recycling, sustaining food webs, and shaping habitat conditions, which collectively enhance ecosystem resilience and productivity. Bacteria and fungi play a crucial role in this process by breaking down coarse particulate organic matter (CPOM), such as leaf litter, into nutrients available for other organisms. However, the specific contribution of bacteria and their functional interactions with fungi in freshwater sediments have yet to be thoroughly explored. In the following study, we enriched organic matter through the addition of alder (*Alnus glutinosa*) leaves into artificial stream channels (AquaFlow mesocosms). We then investigated enzyme expression, metabolic pathways, and community composition of fungi and bacteria involved in the degradation of CPOM through metatranscriptomics and amplicon sequencing. Enzymes involved in the degradation of lignin, cellulose, and hemicellulose were selectively upregulated with increased organic matter. Analysis of ITS and 16S rRNA gene sequences revealed that during decomposition, fungal communities were predominantly composed of *Basidiomycota* and *Ascomycota*, while bacterial communities were largely dominated by *Pseudomonadota* and *Bacteroidota*. The similar gene expression patterns of CPOM degradation related enzymes observed between bacteria and fungi indicate

potential functional interaction between these microbial groups. This correlation in enzyme expression may indicate that bacteria and fungi are jointly involved in the breakdown of coarse particulate organic matter, potentially through mutualistic interaction. This study uncovers the specific enzymatic activities of bacteria and fungi and the importance of microbial interactions in organic matter decomposition, revealing their central role in facilitating nutrient cycling and maintaining the ecological health and stability of freshwater ecosystems.

# INTRODUCTION

Streams receive organic matter from various sources, including wood, leaves, floral parts, and bark (*Benfield, 1997*). Leaf litter, in particular, contributes significantly to the carbon content of the stream food web (*Abelho, 2001*; *Brett et al., 2017*). The decomposition of leaf litter in freshwater streams is a crucial ecological process influencing nutrient cycling, energy flow, and overall ecosystem functioning (*Wallace et al., 1997*). This leaf litter degradation contributes to the formation of coarse particulate, fine particulate, and dissolved organic matter (CPOM, FPOM, and DOM, respectively) by fragmentation and decomposition and thus serves as a vital resource of smaller organic particles and nutrients for microbial communities and aquatic organisms (*Abelho, 2001*; *Graça, Ferreira & Coimbra, 2001*; *Rosi-Marshall et al., 2010*).

The decomposition of autumn-shed leaves follows distinct phases of leaching, physical fragmentation, microbial colonisation, and macroinvertebrate feeding (*Petersen & Cummins, 1974*; *Gessner, Chauvet & Dobson, 1999*). Microbial colonisation occurs as bacteria and fungi attach to the leaf surface and begin breaking down complex organic molecules. This is a key driver for the decomposition of CPOM, organic particles larger than 1 mm (*Webster & Meyer, 1997*). For the deploymerization of leaf litter, microbes such as bacteria and fungi produce extracellular enzymes for the breakdown of leaf cell wall components (*Chróst, 1991*). Aquatic fungi, such as aquatic hyphomycetes, produce enzymes such as pectinase, hemicellulase, and cellulase, (*Zemek et al., 1985*; *Abdel-Raheem & Shearer, 2002*; *Suberkropp & Klug, 1980*). In addition, bacteria also produce cellulolytic and hemicellulolytic enzymes (*Tanaka, 1993*; *Sala & Hans, 2004*). Bacteria and fungi share functions that lead to coexistence as mentioned in *Romaní et al. (2006)* where they point out both synergism and antagonism between bacteria and fungi in leaf litter decomposition. Fungi are more active than bacteria in the starting phase of decomposition leading to the breakdown of cell wall components such as lignin, which makes nutrients accessible to bacteria and leads to higher activity in the later stage of the decomposition (*Suberkropp & Klug, 1976*; *Gessner & Chauvet, 1994*; *Romaní et al., 2006*).

Previous studies on leaf litter decomposition have primarily focused on decomposition rates and fungal community composition (*David et al., 2024*; *Becu & Richardson, 2023*; *Abril, Menéndez & Ferreira, 2021*) often overlooking the complex interactions involving

bacteria. While enzymatic assays (*Smart & Jackson, 2009*; *Zhu et al., 2022*) provide insights into total activity, they typically lack differentiation between bacterial and fungal contributions.

Most research has targeted fungi (*Abril, Menéndez & Ferreira, 2021*), with fewer studies considering bacteria (*Hieber Ruiz & Gessner, 2002*; *Romaní et al., 2006*; *Hayer et al., 2016*; *Wang et al., 2022*). *Wang et al. (2022)* showed with activity assays and metabarcoding that litter input manipulation changes enzyme activity in forest soil by altering the community composition for bacteria and fungi. Metatranscriptome analysis, which enables the distinction between bacterial and fungal functions, has been largely underutilized so far. Species composition and transcriptional activity of the fungal community in terrestrial leaf litter were assessed by metabarcoding and metatranscriptome sequencing by *Guerreiro et al. (2023)*. Here the authors showed functional redundancy of fungal communities with regard to their encoded CAZyme families such as GH9 and PL4, and highlighted the phyla Ascomycota and Basidiomycota as being most abundant. Likewise, freshwater streambed studies mainly focused on fungal communities as investigated by *Cornut et al. (2010)*. The interaction between bacteria and fungi and the dynamics of organic matter degradation by these microbial groups remain poorly understood and are, to our knowledge, not investigated so far in freshwater streambeds.

Therefore, this study aims to investigate the enzymatic expressions and taxonomic composition of fungal and bacterial groups over time as well as their functional relationship in organic matter degradation in freshwater streams. To achieve this, metatranscriptome and amplicon sequencing were utilized to capture transcript expression linked to decomposition enzymes and to analyze community composition throughout the decomposition process. We further applied the classical sporulation method (*Bärlocher, Gessner & Graça, 2020*) using leaf-litter discs, to identify active fungal species colonising degrading leaves. By combining molecular tools with traditional ecological approaches, we aim to contribute to a deeper understanding of the fundamental processes driving leaf litter decomposition in freshwater streams and its implications for ecosystem health and function.

# MATERIALS AND METHODS

## Experimental setup and sampling

The experiment was conducted at the greenhouse of the University of Duisburg Essen from November 11th to 30th where six Aquaflow mesocosms (*Graupner et al., 2017*) were used (Fig. S1). Each system had two large tanks which were connected by two sediment-filled channels (narrow and broad). Each system was filled with 365 L of water and 60 L of sediment from the Boye River (51°33′17.8″N 6°56′43.2″E). Water was pre-filtered with a nylon net bag (200 μm) to keep out metazoa and sediment was mixed with a concrete mixer to homogenize the material. After homogenization, 45 cylindrical sediment cores (Height: 9 cm and Diameter: 3 cm) were placed in the channels of each system in a single row. The remaining portion of the flume was then filled with sediment to a height of 10 cm. Alder (*Alnus glutinosa*) trees were purchased from Eggert Baumschule (Vaale, Germany) and cultivated in the botanical garden of the University of Duisburg-

Essen. Senescent leaves were collected from alder trees in the autumn of 2021. Only leaves were taken that were easily detached because they had already developed a parting tissue, indicating that they were about to be naturally abscised. The collected leaves were subsequently air-dried for a week. Dry leaves were put in small bags of garden net (5 mm). Each bag contained 0.8 g of leaves (~2–3 leaves) without a petiole. The collected leaves were also used to cut leaf-discs (diameter 10 mm) that were placed in fine mesh bags (36 bags, three discs per bag). All leaf bags were incubated with mixed leaves from the Boye River bed before the experiment to ensure colonisation by fungi and bacteria. For incubation, 400 partially decomposed leaves from Boye River such as hazelnut, poplar, birch, and alder were mixed with the alder leaf bags with 180 L of water from the Boye River. Three bubble air pumps were placed for oxygen supply at the bottom of the tanks. Following a seven-day incubation, the leaf litter bags and discs were introduced into the narrow channels of three systems (treatment), while the remaining three systems were left without leaves (control). Each channel contained 90 bags of leaves and six bags of leaf discs, placed above the sediment cores. Two large bags (25 × 20 cm), each containing 25 leaves, were placed in large tanks for initial colonisation in each of the six systems from the start of acclimatisation. These bags were removed after the fourth day of acclimatisation. The water temperature in each channel was set to 15 °C with the help of water-refrigerator TC20 (TECO SRL, Ravenna, ITA). This setup was acclimatised for 10 days by exchanging the water between all six systems daily to obtain the same initial conditions. Following the acclimatisation phase, sediment was systematically sampled for 10 days, whereas the leaf discs were sampled on days 1, 5, and 10. For sediment sampling, four cores were extracted from each flume every day and then homogenized together in a beaker. Subsequently, 2 g of the homogenized sediment was aliquoted into five separate 2 ml Eppendorf tubes. A total of 60 sediment samples were collected. These samples were promptly stored in a cryo-shipper charged with liquid nitrogen for transportation to the laboratory of the University Duisburg-Essen. They were kept at −80 °C until the commencement of nucleotide extraction.

## Nucleic acid extraction and sequencing

To extract RNA from sediment samples, 5 g of sample material was processed as described in *Buchner & Wolany (2023)*. Briefly, samples underwent homogenisation and lysis using an 800 µl guanidine isothiocyanate (GITC) lysis buffer and phenol-chloroform, employing a bead-beating method. RNA precipitation and binding were carried out using silica spin columns. Following thorough washing, RNA was eluted with 100 µl elution buffer at room temperature. The obtained RNA was subsequently subjected to sequencing at CeGaT NovaSeq in Tübingen, Germany. Library preparation and rRNA removal were achieved using Illumina TruSeq Stranded RNA and Ribo-Zero kits. Samples were sequenced with CeGat NovaSeq 6000, 101 bp sequencing (2 × 101).

For amplicon sequencing, DNA was extracted from 500 mg of each sample for bacteria and fungi. A mixture of 0.1 and 0.5 mm diameter glass beads and 100 µl Proteinase K, 5 µl RNAse A, and 900 µl TNES were used for bead beating method with Mini-Bead-Beater 96 (Biospec Products, Bartlesville, OK, USA) for 2 min at 2,400 rpm (for buffer and reagents

see materials in *Buchner (2022a)*. Samples were incubated at 56 °C and bead beaten at 1,400 rpm for 20 min. PCR replicates were divided from lysates and the spin column protocol using a vacuum manifold was used for DNA extraction (*Buchner, 2022b*). DNA clean-up (*Buchner, 2022c*) was performed with 40 μl DNA input and 80 μl of clean-up solution. A two-step PCR approach was used for DNA amplification. For the amplification of bacterial DNA, 515f/806r primers (*Apprill et al., 2015*) were used with a reaction volume of 10 μl, incorporating 1 μl of DNA. Similarly, for fungal DNA, fITS7 (*Ihrmark et al., 2012*) and ITS4 (*White et al., 1990*) primers were used with a 10 μl reaction volume and 1 μl of DNA in the initial PCR (Multiplex PCR Plus Kit; Qiagen, Hilden, Germany). First PCR is followed by the cleanup (*Buchner, 2022c*) and then 2 μl used for the second PCR. The cycling condition for the first and second PCR is given in Table S1. Sample where noramlised to ~2 ng/μl a with bead-based normalisation (*Buchner, 2022d*). For pooled libraries, a Spin-column clean-up protocol was used which resulted in a final volume of 100 μL (*Buchner, 2022b*). Two PCR replicates were produced from each sample to ensure accuracy and reliability. These libraries were then subjected to paired-end sequencing for bacteria (2 × 250 bp) on Illumina NovaSeq and fungi (2 × 300 bp) on Illumina MiSeq V3 (CeGat Gmbh, Tübingen, Germany). Amplicon sequencing data were generated utilizing an established, automated pipeline developed by a large consortium handling numerous samples and projects. In contrast, metatranscriptome sequencing required phenol-chloroform-based extractions to produce sufficient quantities of RNA. This extraction method could not be incorporated into the automated pipeline. However, as both extraction protocols utilize SDS and bead-beating for cell lysis, we consider the resulting data to be comparable.

## Bioinformatic analysis and statistics

### Metatranscriptomics

For metatranscriptome analysis we employed a comprehensive in-house Snakemake (v4.3.1, *Köster & Rahmann, 2012*) workflow (https://github.com/adeep619/Vasuki.git) to process raw total RNA sequences, providing detailed taxonomic and functional annotations. Initially, the raw reads were cleaned to remove adapters and primers using Cutadapt (v4.9, *Marcel, 2011*) with a PHRED score above 20 and subsequently filtered to exclude rRNA sequences with RiboDetector (v0.2.7, *Deng et al., 2022*). To eliminate plant-derived mRNA, the remaining reads were mapped against the *Alnus glutinosa* genome using Bowtie2 (v2.4, *Langmead & Salzberg, 2012*) and Samtools (v1.3.1, *Li et al., 2009*) and filtered out. Non-plant-derived mRNA reads were kept and assembled into contigs using rnaSPAdes (v4.0.0, *Bushmanova et al., 2019*) and quantification was done with salmon (v1.10.3, *Patro et al., 2017*). Both were used with default parameters. These contigs were then subjected to taxonomic annotation; bacterial sequences were identified through searching against the NCBI NR database (filtered for bacterial sequences only by using taxa ID) (v30.06.2023), while fungal sequences were mapped against the JGI Mycocosm database (*Grigoriev et al., 2014*, v30.05.2023), using Diamond BLASTX (v2.0.4, *Buchfink, Xie & Huson, 2015*). The best hit was filtered to include only alignments with an e-value of 1e-10 or lower for both bacteria and fungi. CDS sequences for each genome were

downloaded from the JGI Mycocosm database using an in-house Python script and personal credentials. For functional annotation, all assembled contigs were searched against the UniProt database using Diamond BLASTX (v2.0.4, *Buchfink, Xie & Huson, 2015*) and UniProt accessions were annotated with KEGG Orthologs (KOs) from the KEGG (Kyoto Encyclopedia of Genes and Genomes) database. Custom Python scripts were utilised to merge the taxonomic and functional annotations, providing a comprehensive profile of the mRNA sequences. All diamond BLASTX search was performed with parameters -b 20 (block-size) and–max-target-seqs 20.

Statistical analysis was conducted using R (v4.3.3, *R Core Team, 2024*). The exploration and representation of filtered read count data (removal of KOs with a row sum of less than 100 reads) involved a variance stabilizing transformation and principal component analysis using the variance Stabilizing Transformation function (parameters: blind = TRUE) from DESeq2 (v1.42.1, *Love, Huber & Anders, 2014*) and the R built-in function prcomp for inter-group comparisons. Alpha diversity was assessed over time between groups, and statistical significance was determined using vegan (v2.6-6.1, *Oksanen et al., 2024*) and ANOVA from ggpubr v0.6.0 (*Kassambara, 2023*). Abundance box plots were generated to illustrate pathways associated with CPOM degradation on days with significantly higher functional diversity. For this, pathways related to CPOM degradation were curated from literature (*Datta et al., 2017*; *Schroeter et al., 2022*; *Bonnin & Pelloux, 2020*; *Pollegioni, Tonin & Rosini, 2015*; *Haile & Ayele, 2022*) (Table S3). Temporal analysis of KEGG orthologs over 10 days was done with TCseq (v1.26.0; *Wu & Gu, 2024*) which captures the temporal patterns by unsupervised clustering. For this, the timecourseTable function was used with parameters filter = T, norm.method = rpkm, and value = "expression" to form a time course table of KOs abundance. This table was used for clustering with cmeans and expression values were transformed to Z-scores with parameter standardize = TRUE. A cluster with upregulated pattern of KOs was used for further functional analysis. For differential analysis, DESeq2 (*Love, Huber & Anders, 2014*) was used. KOs abundance was filtered for reads with less than 100 counts. Differential analysis was performed with generalised linear model condition*day to capture gene expression changes due to both, treatment and time. Significant KEGG Orthologs (p_adjusted ≤ 0.05 and log2FoldChage ≥ 2) were annotated to EC numbers and CAZyme families using the KEGG Orthologs database and annotation within (*Kanehisa et al., 2016*; v01.06.2024). MAplots were generated with the R package ggpubr (v0.6.0; *Kassambara, 2018*). For this analysis, significantly differentially expressed KOs (p-adjusted ≤ 0.05 and log2FoldChange ≥ 2) were subjected to KEGG enrichment using the R package gage (v2.52.0; *Luo et al., 2009*) and the kegg.gsets function. The parameters species = "ko" and id.type = "kegg" were specified to ensure accurate annotation and enrichment analysis.

### 16S rRNA and ITS amplicons

Raw amplicon sequences were analysed with the Natrix2 snakemake workflow (*Deep et al., 2023*). Briefly, this workflow includes primer removal, assembly with Pandaseq (v2.11; *Masella et al., 2012*), and filtering the paired-end reads with an alignment threshold score of 0.9 and sequence length with a minimum of 100 bp and a maximum of 600 bp.

Dereplication (100% sequence similarity) and removal of chimeric sequences was done with cd-hit (v 4.8.1; *Fu et al., 2012*); erroneous sequences were removed with a split sample approach using AmpliconDuo (v1.1, *Lange et al., 2015*). Resulting sequences were clustered with Swarm (v2.2.2, *Mahé et al., 2014*) into OTUs which were aligned against the Silva database (v138.1; *Quast et al., 2012*) for bacteria and Unite (*Abarenkov et al., 2024*) for fungi by using Mothur (v1.40.5; *Schloss et al., 2009*) for taxonomic classification with bootstrap value more than 80. MUMU (https://github.com/frederic-mahe/mumu) was used for post-clustering to remove erroneous OTUs. For quality assessment, PCR replicates of each sample were consolidated by eliminating OTUs exclusive to one of the replicates and combining the reads from the remaining replicates. Additionally, the number of reads for OTUs identified in negative controls was subtracted from all individual samples. These OTUs were filtered for row sum greater than 100 for further downstream statistical analysis. Statistical analysis was conducted using R (4.3.3, *R Core Team, 2024*). PhyloSeq (v1.46.0; *McMurdie & Holmes, 2013*) was used for calculation of alpha diversity and visualisation of the community composition with the Bray Curtis distance matrix. DESeq2 (v1.42.1; *Love, Huber & Anders, 2014*) was used to investigate differentially significant abundance changes in taxonomy. The R built-in function prcomp was used for Principal components analysis to detect changes of abundance over treatment and time. R packages such as dplyr (v1.1.4; *Wickham et al., 2023*), tidyverse (v2.0.0, *Wickham et al., 2019*), ggpubr (v0.6.0, *Kassambara, 2018*) microViz (v0.12.1; *Barnett, Arts & Penders, 2021*) were used for data handling and graphical representation.

## RESULTS

### Effect of allochthonous leaf litter on global expression patterns

Raw reads were quality-filtered, assembled, and annotated with functions and taxonomy as mentioned in the material and methods. PCA revealed a slight separation between the control (without leaves) and treatment (with leaves) channels as well as time points for both bacteria and fungi. However, this separation corresponded to a very low explained variance on the higher principal components for treatment (bacteria: PC 5, 3.54%; fungi: PC 6, 3.59%) (Figs. S2 and S3) and for time points (bacteria: PC 4, 4.04%; fungi: PC 4, 4.61%) (Fig. S4). Furthermore, a parametric multivariate statistical permutation test (PERMANOVA conducted using adonis2) indicated no significant differences ($p$-value ≥ 0.05) between the control groups and treatment groups for either bacteria or fungi. Additionally, there were no significant interactions between day and treatment effects (Table S3).

### Effect on functional diversity

The richness and evenness of all KO counts were examined using Shannon diversity. Significant differences between treatment and control samples were observed on Day 4 (ANOVA: $p$-value = 0.042), Day 6 (ANOVA: $p$-value = 0.008), Day 9 (ANOVA: $p$-value = 0.047) for bacteria (Fig. 1A) and Day 4 (ANOVA: $p$-value = 0.03) for fungi (Fig. 1B). While the difference was not statistically significant, functional diversity index was mostly higher for treatment than for control at all time points, especially for fungi

(Fig. 1). A closer look at these latter timepoints showed significantly higher expression of CPOM degradation enzyme in treatment group compared to the control. Notable examples include cellulase, xylanase, pectin lyase, and various glycosidases (Table S5).

## Temporal dynamics of CPOM degrading enzymes

To analyse the temporal change of transcript abundances of enzymes, TCseq analysis was performed for all KOs, and clusters with ascending trends (upregulation over time) were used for further analysis (Fig. 2). For bacterial KOs, these clusters include a group of enzymes involved directly or indirectly in CPOM degradation such as 4-hydroxyphenyl acetate 3-monooxygenase involved in the degradation of aromatic compounds potentially derived from lignin breakdown, Formaldehyde ferredoxin oxidoreductase involved in the oxidation of aldehydes which can be part of lignin breakdown, cellobiosyl-(1->3)-beta-D-glucosidase; involved in the breakdown of cellulose, an important component of plant cell walls, and β-Glucosidase; responsible for breaking down glucosides into glucose, a crucial step in cellulose degradation (Table S7). In fungal KOs, clusters with an upregulation trend include enzymes such as glucuronoxylanase which is involved in the hydrolysis of xylans, a main component of hemicellulose found in plant cell walls and β-glucosidase, an enzyme involved in the hydrolysis of terminal, non-reducing β-D-glucosyl residues with the release of β-D-glucose. It is crucial in the degradation of cellulose into glucose, directly contributing to the degradation of plant matter (Table S7).

For certain genes, bacteria and fungi showed similar mean gene expression ratios over time between treatment and control. Selected KOs specific for CPOM degradation (Table S4) are shown in Fig. 3.

## Differential expression analysis

Differential expression analysis with DESeq2 design 'treatment*day' showed that in bacterial KOs, there were 230 significant KEGG orthologs for the treatment effect (addition of leaf litter), 85 KOs for the time effect, and 102 for the interaction effect of treatment with time (Table S8). Figure 4 illustrates the significant differential expression of enzymes associated with CPOM degradation. Notably, several bacterial enzymes are highlighted, including 3-hydroxybenzoate 6-monooxygenase, which breaks down aromatic compounds from lignin, and sphingolipid C4-monooxygenase, involved in sphingolipid metabolism. Additionally, beta-galactosidase hydrolyzes beta-galactosides into monosaccharides, while mannan endo-1,4-beta-mannosidase contributes to the degradation of the plant cell wall component mannan. Peptidases are also present among other enzymes (Table S8). In addition, CAZyme glycoside hydrolase families such as GH20 and GH26 annotated with significant KOs also showed upregulation with treatment effect (Table S8).

In the case of fungi, 618 KEGG orthologs were significant for the treatment effect, 77 KOs for the time effect, and 96 KOs for the interaction effect of treatment over time (Table S8). Here, the significant KOs annotated to multiple enzymes were involved directly or indirectly in CPOM degradation such as 3-hydroxybenzoate 6-monooxygenase for aromatic compound degradation which are components of lignin, L-xylulose reductase Involved in

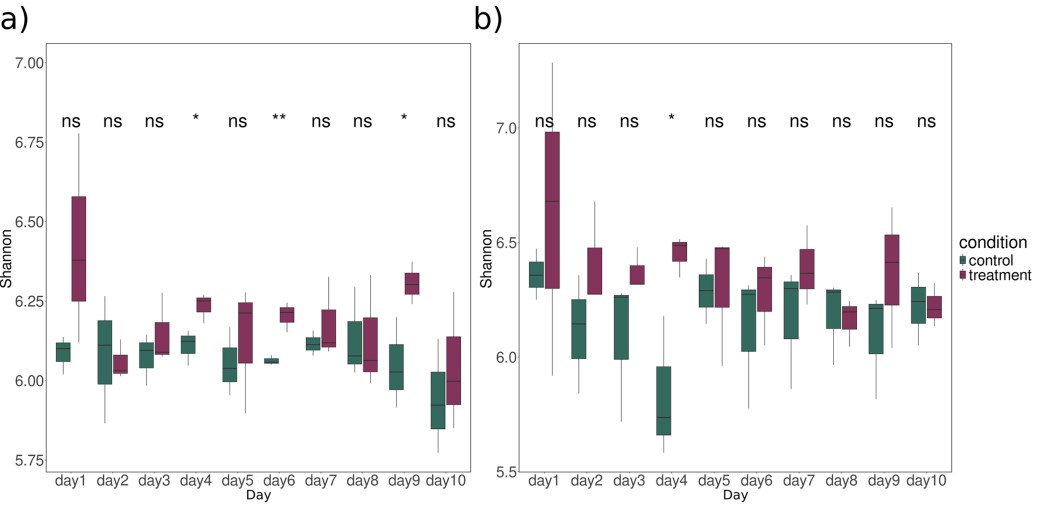

**Figure 1  Alpha diversity of all KO profiles expression over time for (A) bacteria and (B) fungi in sediment samples.** The significance between control (dark green) and treatment (dark raspberry) was tested using ANOVA. Asterisks indicate the significance levels of the ANOVA test with (ns: $p$-value > 0.05, *: $p$-value < 0.05, **: $p$-value < 0.01). The x-axis represents the time period in days, and the y-axis represents the Shannon index.

pentose catabolism, important for the breakdown of hemicelluloses like xylan present in leaf litter, glucarate O-hydroxycinnamoyltransferase catalyzes the transfer of hydroxycinnamoyl groups to glucarate, related to the metabolism of plant-derived acids, Beta-galactosidase hydrolyzes beta-galactosides into monosaccharides, Alpha-mannosidase catalyzes the hydrolysis of alpha-mannosides to release mannose which is important for degrading mannans and glycoproteins in leaf litter are among other enzymes which are involved in carbohydrate metabolism, lipid metabolism, and protein degradation are listed in Table S8. In addition, significant KOs were also annotated to the CAZyme family Glycoside hydrolase: GH26 and GH35. Glycoside hydrolases like GH26 and GH35 are known for their involvement in the degradation of complex polysaccharides into simpler sugars, facilitating carbon release from organic matter. GH26 is typically associated with the hydrolysis of hemicellulose components, whereas GH35 is involved in breaking down Beta-galactosides. We conducted a KEGG pathway enrichment analysis to gain further insights into the pathways of upregulated differentially expressed genes. In the context of degradation, significant KOs ($p$-value < 0.05) in both bacteria and fungi are enriched in crucial pathways including starch and sucrose metabolism, glycan degradation, alanine-aspartate and glutamate metabolism, and other pathways involved in breaking down of complex organic compounds (Table S9).

## Community composition based on amplicon sequencing

Amplicon analysis using the Natrix2 workflow and filtering (count data Table S10) generated 5295 OTUs from total 200 million raw sequences for bacteria and 108 OTUs from total 9 million raw sequences for fungi. To see the effect of treatment in the community composition ordination was performed with PCA showing minor separation between control and treatment for bacteria (principal components accounted for axis 1

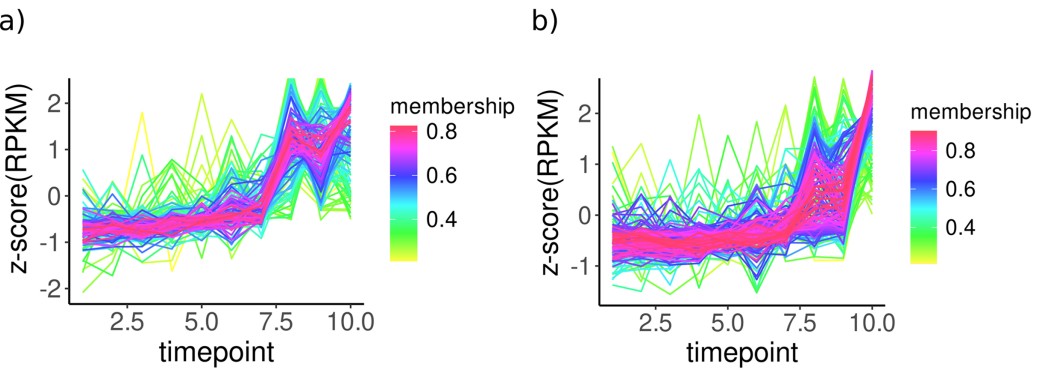

**Figure 2** Temporal patterns for KO expressions of (A) bacteria and (B) fungi from clusters only with ascending expressions in sediment samples. KO expressions were normalised (rpkm) and used to perform cmeans clustering over time. All KOs without changes were removed using filtering within TCseq. The x-axis shows time points over 10 days and the y-axis represents normalised KO expressions (z-scores). Membership values indicate the degree to which data points belong to a cluster.

12.86% and axis 2 7.71% of the total variance) and fungi (principal components accounted for axis 1 22.23% and axis 2 7% of the total variance) (Fig. S5). PC analysis also showed separation of samples with time points at lower axes for bacteria (PC6: 3.71%) and fungi (PC6: 4.33%) (Fig. S6). Observed differences were validated using a PERMANOVA test, which showed that fungi had significant changes in composition between control and treatment while bacteria showed significant differences for condition and day (Table S11). In addition, we observed a significant change in diversity between control and treatment samples for fungi (Fig. S7). A closer look at microbial composition revealed that bacterial communities *Pseudomonadota* and *Bacteriodota* were the most abundant phyla (Fig. 5A) but differential analysis of OTUs abundance on phylum-level showed no significant change with treatment. Fungal communities were predominantly composed of phyla such as *Basidiomycota* and *Ascomycota* (Fig. 5B) but none were related to aquatic hyphomycetes (see Supplemental Information) on the genus level (*Franco-Duarte et al., 2022*) which can be due to primer bias and usage of more universal primers. *Basidiomycota* and *Ascomycota* abundances also show a significantly higher abundance within treatment compared to control (padj < 0.05 & log2foldchange > 2) (Table S12).

# DISCUSSION

This study was set out with the aim of deciphering the functional and taxonomic composition of bacteria and fungi during CPOM degradation in river sediments. Prior studies mostly focused on the microbial communities involved in CPOM degradation of leaves from terrestrial floors, leaf biomass loss in streams, and fungal activity. Use of alder leaves intended to demonstrate comparability with prior studies and to reflect *in situ* dominance around streams (*Abelho, 2001*; *Abril, Menéndez & Ferreira, 2021*; *Graça, Ferreira & Coimbra, 2001*; *Hieber Ruiz & Gessner, 2002*). Our study shows that enzymes like cellulase, pectate lyase and, laccase were expressed by both bacteria and fungi in

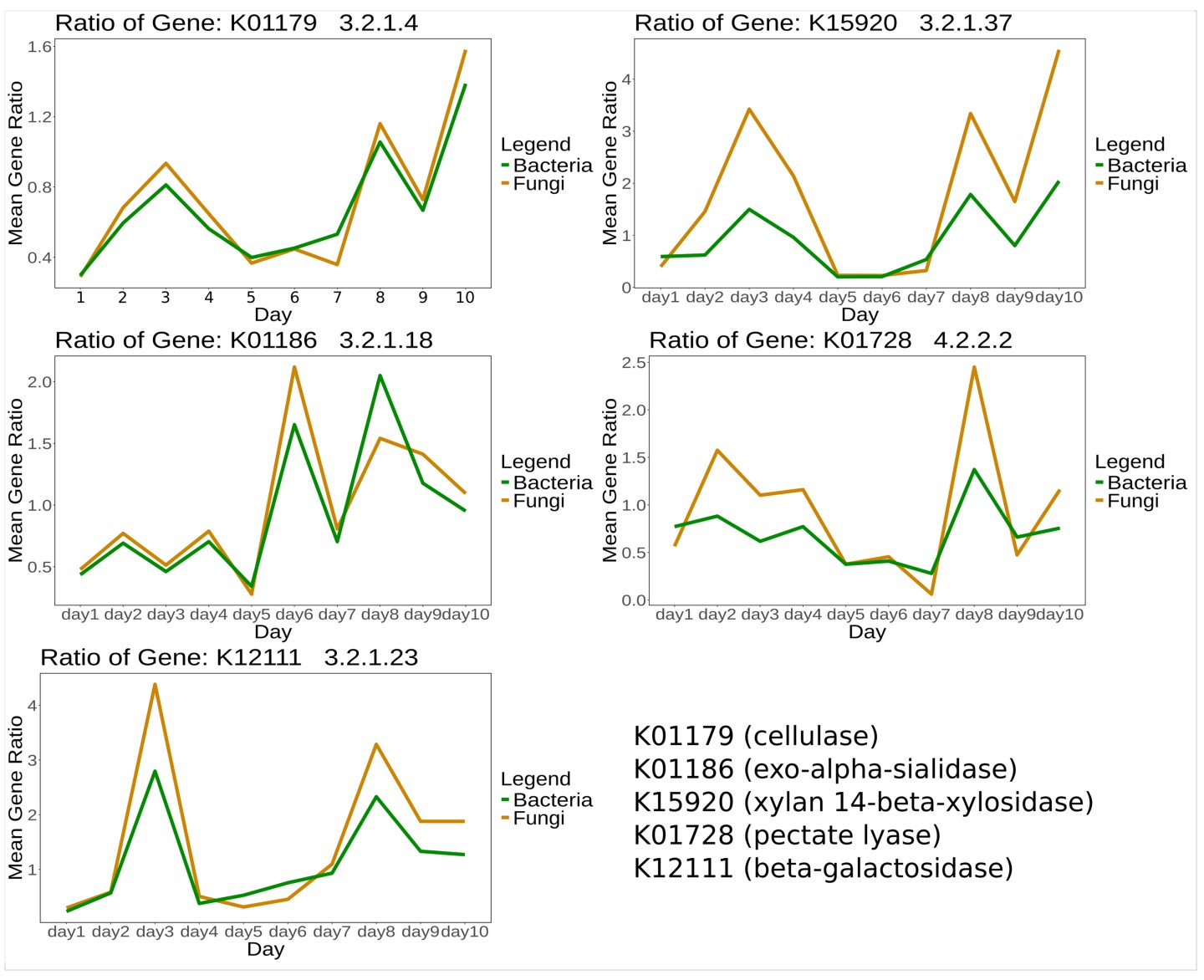

**Figure 3** **Mean expression ratio of CPOM related enzymes such as K01179 (cellulase), K01186 (exo-alpha-sialidase), K15920 (xylan 14-beta-xylosidase), K01728 (pectate lyase) and K12111 (beta-galactosidase) in between control and treatment samples.** The gene expression ratio of listed enzymes was calculated between control and treatment and then the mean expression ratio was plotted for each day.

sediment samples amended with organic matter which aligns with the study by *Wang et al. (2022)*.

## Cooperative functional relationship between bacteria and fungi

Shared temporal expressions of beta-galactosidase, differential expressions of mannan endo14-beta-mannosidase, and KEGG pathways involved in breakdown of complex carbohydrate and aromatic compounds by both bacteria and fungi suggest potential synergism for organic matter degradation in sediment which is in line with the study by *Wohl & McArthur (2001)* in water of freshwater streams. In comparison to bacteria, fungi

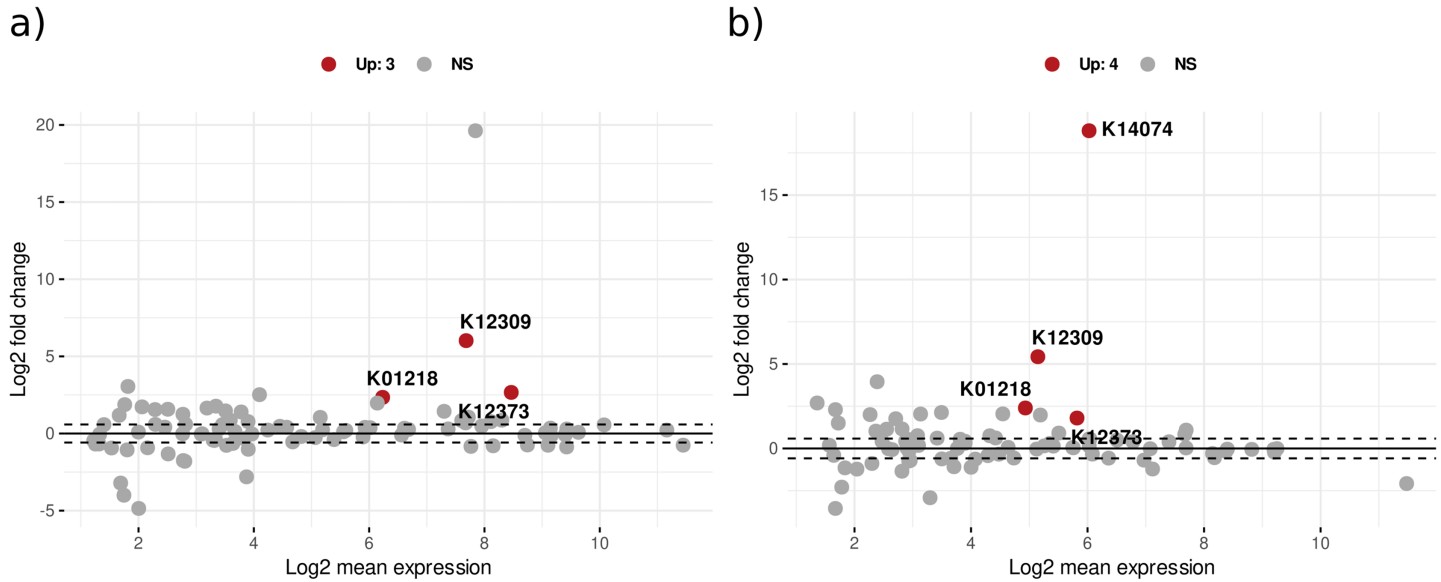

**Figure 4 The MA plot displays the mean expression (A) *vs.* the log-fold change (M) for significant KEGG Orthologs related to CPOM degradation in (A) bacteria and (B) fungi.** The x-axis represents the log2 mean expression and the y-axis indicates the log2 fold change of KOs. A log-fold change greater than 2 and an adjusted *p*-value of 0.05 or above are considered upregulated (Up), whereas values below this threshold are deemed non-significant (NS).

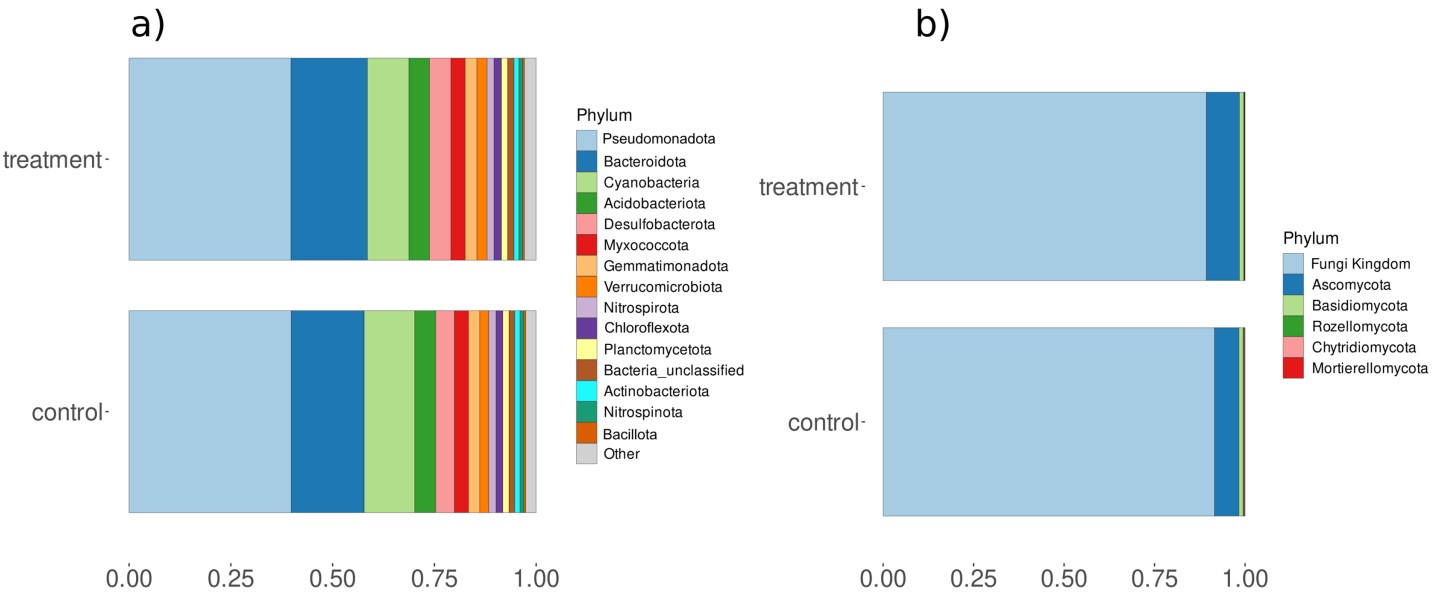

**Figure 5 Relative abundances of (A) bacterial and (B) fungal phyla for control and treatment samples.** The X-axis represents the percent relative abundance of the top 15 phyla, while the Y-axis indicates the conditions (control and treatment).

show more differentially expressed enzymes related to the breakdown of complex carbohydrates with the increase of organic matter which correlates to more fungal activity over time. In comparison to bacteria, fungi show more differentially expressed enzymes related to the breakdown of complex carbohydrates with the increase of organic matter which correlates to more fungal activity over time due to higher expression or higher

fungal abundance (*Gulis & Suberkropp, 2003*; *Romaní et al., 2006*). The observed higher fungal activity aligns with their recognized primary role in the early stages of decomposition, where they initiate the breakdown of complex organic matter, making it more accessible for subsequent microbial processing.

### Consistent bacteria amidst diverse fungi

The analysis of community composition using 16S rRNA gene sequences showed a stable bacterial composition with phyla such as *Pseudomonadota*, and *Bacteriodota*, being the most abundant. These are known to play a crucial role in the degradation of coarse particulate organic matter (CPOM) as mentioned in previous studies (*Newton et al., 2011*; *Besemer et al., 2012*; *Purahong et al., 2016*; *Hayer et al., 2016*). Similarly, the analysis using the ITS region reveals the presence of fungal phyla such as *Basidiomycota*, and *Ascomycota* which also showed significantly higher abundance in differential analysis. These fungal groups are known for their relevance in decomposition processes, as documented in studies by *Meng et al. (2024)*, *Lundell, Makela & Hilden (2010)*, and *Kuramae et al. (2013)*. However, none of the phyla detected belong to a known aquatic hyphomycetes species (*Franco-Duarte et al., 2022*). The absence of this important group of freshwater decomposers can be experimentally explained by the physico-chemical conditions in the sediments, such as oxygen depletion and low turbulence (*Chauvet et al., 2016*; *Cornut et al., 2014*), usage of universal primer due to varying sequence lengths in hyphomycete's ITS regions or potentially missing references of undetermined hyphomycetes in the Unite database.

However, we did not see significant changes for all days (PERMANOVA test) but we did observe significant changes in specific enzymes (inferred from DESeq2) and diversity and richness of CPOM-related enzymes for certain days (Fig. 1). Limitations of our studies could be a shorter duration of the study compared to other traditional experiments, as noted in studies like *Hättenschwiler, Tiunov & Scheu (2005)* and a long period of decomposition process by microbes (90 days) as mentioned in *Petersen & Cummins (1974)*. Furthermore, the most important part of the CPOM decomposition occurs directly on the leaf litter (*Cornut et al., 2014*) which could not be comparatively assessed in flumes without leaves. Despite the different significance levels for various enzymes, the overall expression patterns of CPOM related KOs in both fungi and bacteria display a similar trend over time (Fig. 3). This similarity may indicate potential synergism which is also mentioned by *Bengtsson (1992)* and functional redundancy.

## CONCLUSIONS

While fungal community composition shifted with amended organic matter, both bacteria and fungi actively contributed to organic matter degradation, as evidenced by significant gene expression changes. Despite the stability of the bacterial composition, the shared KEGG pathways, gene expression patterns, and upregulated CPOM-related enzymes between bacteria and fungi suggest a potential functional synergy and redundancy between these microbial groups in the decomposition process.

## ACKNOWLEDGEMENTS

The authors extend their thanks to all participating student assistants, and support staff for their contributions to the AquaFlow setup and sampling efforts. We also thank Julia Engelmann for the help with the statistical analysis. I acknowledge the use of ChatGPT. It was used to improve grammar and readability of the manuscript. Following the use of this tool, the authors revised the text and take full responsibility for the content of this publication.

### Funding

This study was performed as part of the Collaborative Research Center (CRC) RESIST and analyses were performed by Project A04 (AD and DB), funded by the German Research Foundation (DFG)—CRC 1439/1; project number 426547801. This was supported by the University of Duisburg-Essen's Open Access Publication Fund. The funders had no role in study design, data collection and analysis, decision to publish, or preparation of the manuscript.

### Grant Disclosures

The following grant information was disclosed by the authors:
Collaborative Research Center (CRC).
RESIST and analyses were performed: A04 (AD and DB).
German Research Foundation (DFG): CRC 1439/1; project number 426547801.
University of Duisburg-Essen's Open Access Publication Fund.

### Competing Interests

The authors declare that they have no competing interests.

### Author Contributions

- Aman Deep performed the experiments, analyzed the data, prepared figures and/or tables, authored or reviewed drafts of the article, and approved the final draft.
- Guido Sieber performed the experiments, authored or reviewed drafts of the article, and approved the final draft.
- Lisa Boden performed the experiments, authored or reviewed drafts of the article, and approved the final draft.
- Gwendoline M. David performed the experiments, analyzed the data, prepared figures and/or tables, authored or reviewed drafts of the article, and approved the final draft.
- Daria Baikova performed the experiments, authored or reviewed drafts of the article, and approved the final draft.
- Dominik Buchner performed the experiments, authored or reviewed drafts of the article, and approved the final draft.
- Jörn Starke performed the experiments, authored or reviewed drafts of the article, and approved the final draft.

- Tom L. Stach performed the experiments, authored or reviewed drafts of the article, and approved the final draft.
- Torben Reinders performed the experiments, authored or reviewed drafts of the article, and approved the final draft.
- Una Hadžiomerović performed the experiments, authored or reviewed drafts of the article, and approved the final draft.
- Sára Beszteri performed the experiments, authored or reviewed drafts of the article, and approved the final draft.
- Alexander J. Probst conceived and designed the experiments, authored or reviewed drafts of the article, and approved the final draft.
- Jens Boenigk conceived and designed the experiments, authored or reviewed drafts of the article, and approved the final draft.
- Daniela Beisser conceived and designed the experiments, analyzed the data, prepared figures and/or tables, authored or reviewed drafts of the article, and approved the final draft.

## Data Availability

The sequencing data is available at NCBI: PRJNA1164467.

## Supplemental Information

Supplemental information for this article can be found online at http://dx.doi.org/10.7717/peerj.19120#supplemental-information.

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
