# Peer review of "A metatranscriptomic exploration of fungal and bacterial contributions to allochthonous leaf litter decomposition in the streambed"

_PeerJ, doi:10.7717/peerj.19120_

## Round 0.1 · original submission · Minor Revisions

Dear Authors,

Thank you for submitting your manuscript, "A Metatranscriptomic Exploration of Fungal and Bacterial Contributions to Leaf Litter Decomposition in the Streambed,". After a thorough review process, we have determined that the manuscript is suitable for minor revisions before it can be considered for publication.

We believe that these revisions will strengthen your manuscript and improve its impact. Please address each point raised by the reviewers and provide a detailed response explaining how you have incorporated their suggestions.

We look forward to receiving your revised submission.

Best regards,

Armando Sunny

Reviewer 1 ·

Basic reporting

no comment

Experimental design

no comment

Validity of the findings

no comment

Additional comments

This study investigates the role of bacteria and fungi in the decomposition of coarse particulate organic matter (CPOM) in freshwater ecosystems, focusing on enzyme expression and microbial community composition. It highlights the functional interactions between bacteria and fungi in nutrient cycling, emphasizing their central role in maintaining ecosystem health and stability. However, there are still some problems:
1. The PCA analysis showed a slight separation between the control and treatment groups, but the explained variance was low. Have you considered further dimensionality reduction or using other methods (such as t-SNE or UMAP) to enhance the interpretability of the data?
2. In terms of functional diversity, although significant differences were observed at some time points, the differences between microbial communities were smaller in the treatment group, which tended to have higher functional diversity across all time points. I wonder if there are patterns related to environmental factors when comparing changes across different time points (e.g., Day 4, Day 6, Day 9).
3. In terms of CPOM degradation enzyme expression, certain genes in both bacteria and fungi showed an upregulation trend in the treatment group. Have you considered the variation of these enzymes across a broader geographic area? Could this reveal broader ecological mechanisms? Regarding the upregulation of CPOM degradation enzymes at different time points, have you considered the impact of other environmental factors, such as temperature or water salinity?
Line 276-277: This sentence seems to have a problem, it is suggested to change to “A closer look at these later time points showed significantly higher expression of CPOM degradation enzymes in the treatment group compared to the control.”

Line 295-297: It is suggested that the sentence be changed to: “For certain genes, bacteria and fungi showed similar mean gene expression ratios over time between treatment and control.”

I could not find Tables 2, 8, 9, and 10 in the supplementary materials.

·

Basic reporting

The paper "A metatranscriptomic exploration of fungal and bacterial contributions to leaf litter decomposition in the streambed" is a novelty and well-conducted study for understand the contribution of autochthonous leaf litter on microbial-mediated leaves decomposition using metatranscriptomic approach. The title and objetive are inconsistent with the experimental desing because non includes the autochthonous leaf litter effect. Is neccesary clarify some concepts and add some experimental specifications for readers clarity (see recomendations). The current version lacks hypothesis that is neccesary include it.

Experimental design

The experimental desing is intereting and innovative. Is neccesary specify if leaves employed are green or senescent; it is critical to understanding enzymatic activity and microbial community structure because N and P concentration changes between green or brown leaves.

Validity of the findings

No comment

Additional comments

Title

I suggest changing the title to “A metatranscriptomic exploration of autochthonous leaf litter influence on fungal and bacterial enzymatic contributions to leaves decomposition in the streambed.”

L99-101 I think that the aim is partially disconnected, I suggest changing it to “investigate the autochthonous leaf litter influence on enzymatic expressions and taxonomic composition of fungal and bacterial groups along leaves decomposition in freshwater streams.”

L119 Please before defining the systems.

L54 I suggest changing “degradation” to “decomposition” when arguing the general process of decay of organic residues until inorganic compounds are released. (mineralization).

L54 I suggest changing “degradation” to “fragmentation” when arguing the process that reduces organic tissues to coarse and dissolved organic particles.

L58-59 This idea is confusing because it isn’t clear if the subject is decomposition (inorganic nutrients releasing) or fragmentation (production of organic particles as heterotrophic resources).

L67-68 I suggest using “depolymerization” when arguing the process that reduces macromolecules (as the cell-wall components) to low-molecular-weight organic molecules assimilable by microorganisms.

L127 Are the leaves collected green or senescent? This specification is critical to understanding enzymatic activity and microbial community structure because N and P concentration changes between green or brown leaves. Please make this specification and argue because use one or the other.

L256 I suggest modifying to “Effect of autochthonous leaf litter on global expression patterns”

L257-261 This clarification isn’t necessary. Please add L58-61 to M&M section.

L270 and L298. I suggest deleting it because subtitles aren’t necessary.

L279-297 Why is independent of litter addition treatment?

L352-353 Please add the direction of this statistical difference (treatment).

Discussion

Is necessary to contextualize the relevance of microbial metatranscriptomic analysis in decomposition studies (terrestrial and aquatic) and identify the experimental conditions mainly the selection of raw material: a) green or brown leaves, b) Why only one Alder species and their relevance in their system (in situ litterfall dominance)

Reviewer 3 ·

Basic reporting

.

Experimental design

.

Validity of the findings

.

Additional comments

Overall, this article is well-written and the study solid. However, it appears that classic microbial ecology tools have not been used and are instead replaced by metrics or approaches that seem less relevant.
However, I was somewhat disappointed by the underutilization of the data, being reduced to the identification of a few groups of genes with differential abundances. Key aspects, such as the specific expressions by certain taxa or the number of distinct proteins they express… are not explored in depth. These are just a few examples of data points that could be leveraged with the author’s data to truly address the question of synergies and complementarities.

I see no strong reason to reject this article, but regret that authors surely could have done much more and better with their data. There is however some major points to address before publication.

Line 45: this is completely expected, as they represent the vast majority of fungi.
Line 46: Similar in what way? How does this suggest an interaction? Even if it’s the summary, this needs to be clarified.
Introduction: Well-written and clearly presents the context and objectives of the study. No corrections to suggest.
Nucleic Acid Extraction and Sequencing: Why is the lysis method different between the RNA and DNA amplicon protocols? Different treatments can introduce extraction biases, which should be discussed when analyzing the results (in addition to amplification biases, especially in fungi, where primers amplify Asco/Basidio-mycota well but poorly amplify other phyla).
Line 195: How was Mycocosm retrieved? What date, what genome files? Accessible via GitHub scripts but should be minimally explained in M&M. Note: several GitHub scripts contain hardcoded non-encrypted login/password combinations.
Line 196: How are cases handled where there’s a hit on both Mycocosm and NCBI NR, but they don’t match?
Line 223: CAZyme family annotation: how, with which reference (and date) or tool?
Figure S4: Errors in the legend (refers to PC5 and PC6, which don’t appear in the figure).
Line 274: If it is not significant, it is not higher.
Lines 279-294: This paragraph seems very “cherry-picking.” At least provide the number of selected genes and the proportion related to CPOM degradation compared to the proportion among non-selected genes.
Lines 299-310 & 313-323: The sentences are too long, and the sequence of enzyme and substrate names makes these paragraphs very difficult to read without understanding their relevance or importance. More synthetic tables (grouped by major substrate types, e.g., lignin/cellulose/starch) would be more appropriate.
Lines 324-329: This point deserves to be more detailed, with at least a full paragraph. The previous ones are only descriptive and very partial, whereas this one seems much more relevant but is too underdeveloped.
Lines 331-342: PCA and Euclidean distance are not well-suited for highly compositional count data like metabarcoding. It would be more relevant to use MDS/NMDS with standard microbial ecology distances (Bray-Curtis and/or UniFrac).
Line 341 / Figure S8: "Mean abundance" is a very unusual measure in microbial ecology. What about more common diversity indices?
Line 350: Are you sure the selected primers properly amplify Hyphomycetes? And is the ITS sequence length of Hyphomycetes compatible with the sequencing performed? Classical fungal primers are truly universal only for Asco/Basidiomycota, and ITS length is highly variable, sometimes too long in certain groups to concatenate paired reads without a suitable metabarcoding workflow.

Line 366: “Shared”: Why refer to synergy and not competition?
Line 370: More fungal activity or more fungal biomass? Is it related to expression changes or population changes?
Line 370: “This is coherent”: Please detail why and how.
Line 380: This contradicts the hypothesis in line 370, where their role was to degrade fungal products.
Line 390: Missing references for metatranscriptomics, and perhaps primers or ITS length unsuitable for metabarcoding.
Line 408: “As evidenced by significant gene expression change”: Why wouldn’t continuous expression also indicate active contribution to degradation?
Line 411: “Synergy” seems like an overinterpretation to me.

---

## Round 0.2 · Minor Revisions

Dear Authors,

After the review process, your manuscript requires minor revisions before it can be accepted for publication. Only one reviewer has requested three specific changes, and once these are addressed, the manuscript will be ready for acceptance.

We greatly appreciate your effort and dedication in preparing this interesting manuscript.

Best regards,
Armando Sunny

·

Basic reporting

no comment

Experimental design

no comment

Validity of the findings

no comment

Additional comments

no comment

Reviewer 3 ·

Basic reporting

The authors have responded to most of my comments, although they should still incorporate some modifications.

I insist that the Materials and Methods section include details on how sequences matching both NCBI and MycoCosm were handled. What metric was used to determine the best hit between MycoCosm and NCBI? E-value, bitscore?

Another essential issue that still needs to be resolved concerns Figure S8. I understand the interest in the average abundance of individual OTUs, but this figure appears to represent the average abundance of all OTUs, which does not make sense to me. Furthermore, the text referring to Figure S8 mentions significant differences between the control and treatment groups, but this figure does not show any such differences. I suggest simply removing this figure along with the corresponding text.

Additionally, I still find it unclear which data were used for the PERMANOVA analysis in Table S11. Was it the raw OTU abundance table or the means? What distance metric was used for PERMANOVA? Euclidean, Bray-Curtis?

Once these comments are addressed, I consider this article valid for publication. However, I remain somewhat disappointed by the underutilization of the authors' data, as I believe they could have done much more—and better—with their data.

Experimental design

See above comments on Figure S8 and Table S11.

Validity of the findings

No comment

---

## Round 0.3 · accepted · Accept

Dear Dr. Deep,

Thank you for your submission to PeerJ.

I am writing to inform you that your manuscript - A metatranscriptomic exploration of fungal and bacterial contributions to allochthonous leaf litter decomposition in the streambed - has been Accepted for publication. Congratulations!

Best regards,
Armando Sunny